# $E^3$Gen: Efficient, Expressive and Editable Avatars Generation

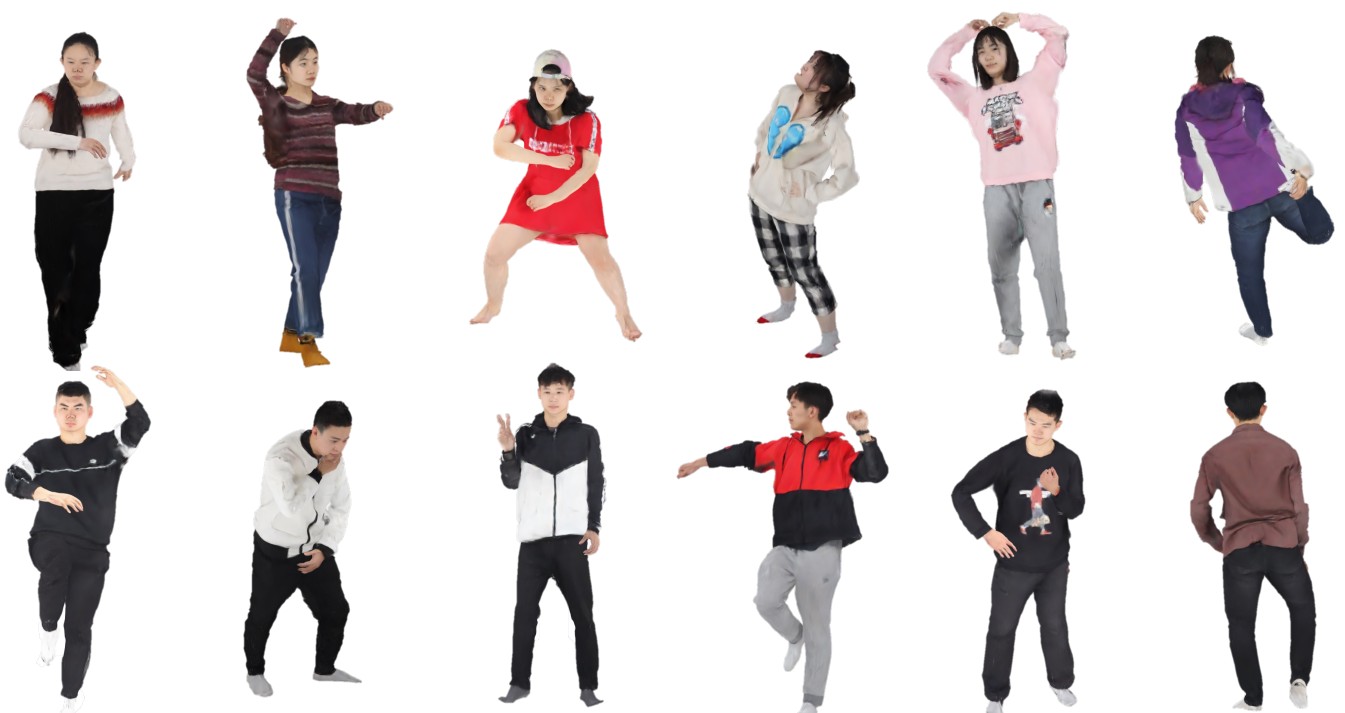

**Figure 1: Our novel method, $E^3$Gen, showcases its capability to generate high-quality animatable avatars with complex textures, providing diverse camera and full-body pose control.**

## ABSTRACT

This paper aims to introduce 3D Gaussians for efficient, expressive, and editable digital avatar generation. This task faces two major challenges: 1) The unstructured nature of 3D Gaussians makes it incompatible with current generation pipelines; 2) the animation of 3D Gaussians in a generative setting that involves training with multiple subjects remains unexplored. In this paper, we propose a novel avatar generation method named $E^3$Gen, to effectively address these challenges. First, we propose a novel generative UV features representation that encodes unstructured 3D Gaussians onto a structured 2D UV space defined by the SMPL-X parametric model. This novel representation not only preserves the representation ability of the original 3D Gaussians but also introduces a shared structure among subjects to enable generative learning of

the diffusion model. To tackle the second challenge, we propose a part-aware deformation module to achieve robust and accurate full-body expressive pose control. Extensive experiments demonstrate that our method achieves superior performance in avatar generation and enables expressive full-body pose control and editing.

## CCS CONCEPTS

• **Do Not Use This Code → Generate the Correct Terms for Your Paper**; *Generate the Correct Terms for Your Paper*; Generate the Correct Terms for Your Paper; Generate the Correct Terms for Your Paper.

## KEYWORDS

Do, Not, Us, This, Code, Put, the, Correct, Terms, for, Your, Paper

**Unpublished working draft. Not for distribution.**

## 1 INTRODUCTION

Digital avatars, we refer to as 3D clothed human characters, have extensive applications [2, 12] in various fields such as virtual and augmented reality, film making, telecommunication, and more. Traditional graphics-based pipelines require weeks of labor from experienced 3D artists, utilizing sophisticated equipment [13, 21, 57] and software, to construct a single digital avatar. This manual and time-consuming process poses a significant obstacle to the creation

of digital avatars at scale. Consequently, there is a pressing need for efficient methods that can generate digital avatars in a fast and autonomous way.

To enable the aforementioned applications, it would be desirable for digital avatars to meet the $E^3$ standard: (1) **Efficient**, *i.e.*, digital avatars are expected to enable real-time, high-resolution, and realistic rendering. (2) **Expressive**, *i.e.*, the avatars should allow animation not only by global body poses but also by local facial expressions and hand gestures. (3) **Editable**, *i.e.*, digital avatars should support easy editing, including local geometry/texture editing and partial attribute transfer, as shown in Fig 1. The key to achieving the $E^3$ standard is to design a powerful generative representation that can cover all these features.

Implicit and explicit representations are the main streams to model digital avatars. Implicit-function-based representations [20, 42] such as neural radiance field [38] can achieve photorealistic rendering results. However, due to the reliance on computationally expensive and time-consuming volume rendering, this representation struggles to support high-resolution and real-time rendering which is crucial in practical applications. The editing ability is also constrained due to the entanglement of geometry and texture. Explicit-mesh-based representations [14, 18, 49], on the other hand, enable high-resolution and real-time rendering through rasterization-based renderer [32] while struggling to represent thin structures like hair which alleviates the realism. A promising alternative for avatar representation is 3D Gaussian [30], which offers realistic rendering quality and real-time high-resolution rendering.

With the flourishing of 3D generative models [4, 17, 18, 26, 35, 39, 50, 55, 61], implicit representations [9, 23, 31, 40, 58] have been widely applied to generate digital avatars. 3D-aware GANs [5] have been applied to generate avatars in canonical pose space [3, 7, 16, 62], followed by a deformation module to transform the avatars to various body poses. These methods can generate avatars under the control of camera viewpoint and body pose, but due to the diverse topology and complex texture of avatars, they cannot realistic rendering results. Recently, diffusion-based methods [15, 22, 46, 54] have surpassed GAN-based methods [27–29] in 2D generation tasks, achieving high-quality generation results in complex and diverse scenes, promising for utilization in 3D content generation. Several attempts [11, 24] have been made to introduce diffusion-based methods for animatable avatar generation. However, the performance is still restricted by the representation of avatars.

It is natural to ask a question: *Can we marry the generation power of diffusion model with the representation ability of 3D Gaussian to achieve the $E^3$ standard?* Our answer is YES, but this is a non-trivial task due to the following reasons. 1) The unstructured nature of the 3D Gaussian poses challenges for its integration into diffusion-based generation pipelines which primarily consists of 2D-CNN-based networks [47]. 2) it is an unexplored problem for the animation of 3D Gaussians in a generative setting.

To address the first challenge, our key insight is to project the unstructured 3D Gaussian into a structured 2D space. Following this idea, we propose a novel avatar representation, *i.e.*, generative UV feature planes. Specifically, we employ the parametric human model SMPL-X [43] as an initial template to provide a shared structure for the human body, and the corresponding 2D UV map is utilized to encode the attributes of 3D Gaussian. We assign the initial position of each 3D Gaussian on the surface of a densified SMPL-X mesh. By decoding the UV feature maps, we obtain the attribute UV maps of 3D Gaussian. Each pixel within these maps represents the attributes of the corresponding 3D Gaussian anchored on the SMPL-X surface. This novel representation offers several benefits. First, it preserves the efficient advantage of the original 3D Gaussians representation while obtaining a shared structure UV space. Second, by utilizing the semantic information of SMPL-X, this representation supports editing. Third, it is compatible with generative models, enabling an efficient and editable avatar generation process.

The remaining question is how to enable expressive body control. Compared to body pose animation, animating the hands and face region is challenging because the small volume and complex deformations of these regions make it difficult to learn or query accurate skinning weights. We observe that hands and face regions have limited topology changes, while body parts undergo large topology changes. Therefore, we develop a part-aware deformation module to enable expressive full-body pose control. Thanks to the generative UV feature plane representation, we can easily assign the accurate blendshapes and skinning weights of SMPL-X for 3D Gaussian in the face and hand region according to their barycentric coordinate. The body parts are animated with precomputed KNN-based skinning weights. Unlike previous methods [59], we utilize a forward skinning scheme [8, 10] instead of an inverse skinning technique which would lead to errors for points in the interaction area. Our novel representation that provides 3D Gaussian distributed over the avatar's surface also facilitates accurate and robust animation, compared to implicit representations that contain many points far away from the geometric surface where blending weights are difficult to assign.

Extensive experiments demonstrate that our method $E^3$Gen achieves superior efficiency, generation quality, and control ability compared to current state-of-the-art methods. Ablation studies have been conducted to evaluate the design choices of our method. Furthermore, $E^3$Gen supports local editing tasks, including local attribute transfer (such as face shape and clothing) and texture editing. In summary, our method has the following contributions:

- We propose $E^3$Gen, an efficient avatar generation pipeline that enables real-time rendering at high resolution (1024 × 1024). It also supports expressive pose control and editing capability.
- We propose a novel 3D avatar presentation called generative UV features to marry the unstructured 3D Gaussians with the current diffusion generation pipeline.
- Our $E^3$Gen achieves superior generation quality on THuman2.0 Dataset and supports local editing including attribute transfer and texture editing.

## 2 RELATED WORK

### 2.1 3D Digital Human Generation

In the field of 3D digital avatar generation, various approaches [3, 9, 58] have been explored to create animatable avatars by combining 3D-Aware GANs [4, 5, 50] or diffusion models [54] with implicit human representations. ENARF-GAN [40] utilizes an articulated neural representation based on tri-planes [5] but struggles to produce high-quality generation results. Other approaches [3, 16, 62]

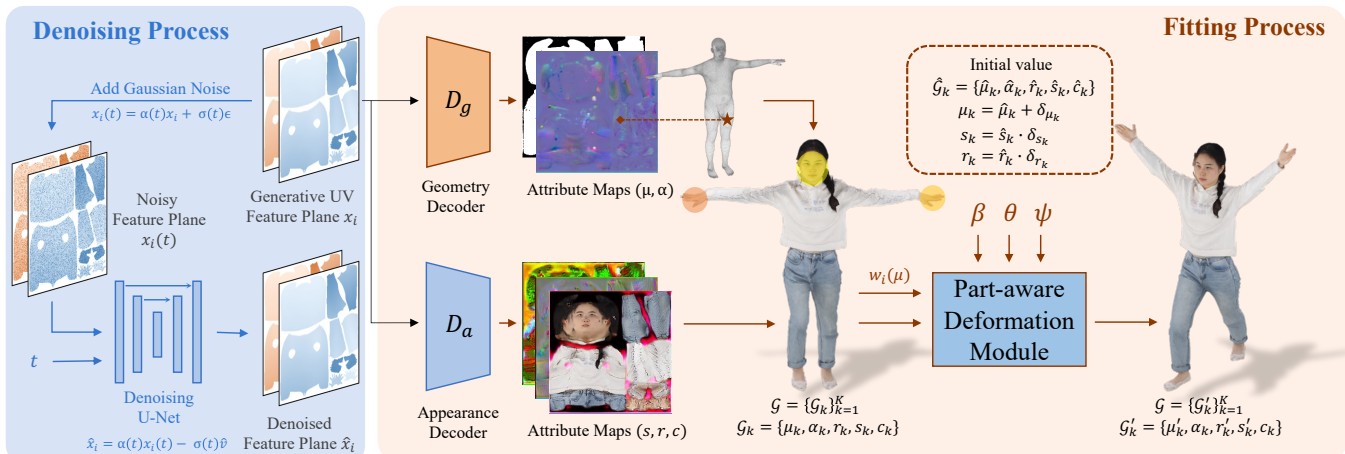

**Figure 2: Method Overview.** Our approach utilizes a single-stage diffusion model to simultaneously train the denoising process and fitting process. The UV features plane, $x_i$, is randomly initialized and optimized by both processes. In the denoising process, noise is added to the UV features plane and then denoised following a v-parameterization scheme using a denoising UNet. In the fitting process, the UV features plane is decoded into Gaussian Attribute maps, which are used to generate a 3D-Gaussian-based avatar in canonical space by fetching the corresponding attributes for the initialized Gaussian primitive. Finally, a part-aware deformation module is employed to deform the avatar into the desired pose based on SMPL-X parameters.

build upon EG3D [5] and employ super-resolution module to enhance the resolution of the generated avatars. Unfortunately, these methods are often hindered by view inconsistency issues, which affect the overall quality of generated avatars. To address this challenge, EVA3D [23] represents the digital human as a compositional part-based human representation. This approach achieves impressive rendering results with resolutions of 512. However, it falls short in enabling real-time rendering and faces difficulties in editing due to the entanglement of geometry and texture. Implicit human representations mainly rely on inverse skinning technique for avatar animation. However, this technique has certain drawbacks. Firstly, it tends to produce artifacts in joint regions and areas of contact. Secondly, for learnable skinning weights fields, it lacks generalization capabilities. To solve these issues, AG3D [16] adopts a forward-skinning technique [10] but faces the costly process of root finding. Another category of methods [18, 19] attempt to incorporate explicit mesh models to achieve real-time high-resolution rendering while providing better support for editing and animation. These mesh-based methods often rely on parametric models such as SMPL [37] to learn offsets or employ neural networks [51] for mesh optimization. However, meshes representations face challenges in accurately representing thin structures, such as hair, and may be constrained by the topology limitations imposed by models like SMPL, resulting in generated avatars that lack realism in their rendering results.

Recently, 3D Gaussians [30] serve as an explicit representation that supports both editing and animation capabilities while enabling real-time high-resolution rendering. This representation offers greater representation capability compared to meshes. A concurrent work, GSM [1], combines 3D Gaussians with EG3D [5] using shell maps [45]. In comparison, our generative UV features plane representation facilitate expressive animation that includes

facial expressions and gestures, while also supporting editing capabilities. Additionally, GSM [1] control body-only poses and faces challenges in editing local regions of the avatar. We do not compare to it since the training code have not been released.

## 2.2 Diffusion Model

The diffusion model has recently achieved significant success in the field of generation, surpassing GANs in tasks such as text-to-image generation. Consequently, there is a growing interest in extending the success of the diffusion model from 2D to 3D generation. One approach [44, 56] leverage prior knowledge encoded in pre-trained latent diffusion models for text-to-3D generation. However, these methods often employ per-subject optimization, which can take hours to generate a single sample, thus reducing efficiency.

Another approach [41, 52] focuses on directly generating 3D representations, enabling faster inference. However, many of these methods adopt a two-stage training scheme. This can introduce noisy patterns and artifacts in the latent code due to the uncertain nature of inverse rendering. Consequently, these noisy patterns can distract denoising networks and affect the quality of the generated outputs. To solve this, SSDNeRF [6] proposes a single-stage diffusion model that trains the fitting and denoising parts together, leveraging the diffusion priors to constrain the latent codes. However, SSDNeRF focuses on static object generation and adopts an implicit representation, which limits the rendering resolution to only 128x128. In comparison, our proposed method can achieve high-resolution rendering at 1024x1024 in real time while generating articulated digital avatars.

## 3 PRELIMINARY

In this section, we provide a brief introduction about SMPL-X human model in Sec. 3.1 and 3D Gaussians Splatting in Sec. 3.2.

## 3.1 SMPL-X

SMPL-X is an animatable parametric human model that represents human body (without cloth) with a parameterized deformable mesh $M(\beta, \theta, \psi)$. This model consists of 10,475 vertices and 54 joints, allowing for control over hand gestures and facial expressions. The deformation process can be formulated as follows:

$$M(\beta, \theta, \psi) = LBS(T_P(\beta, \theta, \psi)), J(\beta), \theta, \mathcal{W}), \qquad (1)$$

where $\beta$, $\theta$ and $\psi$ represent shape, pose and expression parameters respectively. The linear blend skinning (LBS) function, denoted as $LBS(\cdot)$, is used to transform the canonical template $T_P$ the given pose $\theta$ based on the skinning weights $\mathcal{W}$ and joint locations $J(\beta)$. The canonical template $T_P$ can be computed as:

$$T_P(\beta, \theta, \psi) = T_C + B_S(\beta; \mathcal{S}) + B_E(\psi; \epsilon) + B_P(\theta; \mathcal{P}), \qquad (2)$$

where, $T_C$ denotes the mean shape template. $B_S(\beta; \mathcal{S})$, $B_E(\psi; \epsilon)$ and $B_P(\theta; \mathcal{P})$ represent per-vertex displacements calculated by the blend shapes $\mathcal{S}$, $\mathcal{P}$ and $\mathcal{E}$ with their corresponding shape, pose and expression parameters.

## 3.2 Gaussian Splatting

3D Gaussian splatting is an explicit point-based representation for 3D static scenes, involving a collection of 3D Gaussian primitives denoted as $\mathcal{G}$. These primitives enable real-time rendering through differentiable rasterization. Each 3D Gaussian $\mathcal{G}_k$ comprises five attributes: position $\mu$, scaling matrix $S$, rotation matrix $R$, opacity $\alpha$, and view-dependent color $c$, which is represented by coefficients of spherical harmonics. In practice, we employ RGB color instead of spherical harmonics coefficients for simplicity and utilize the diagonal vector $\mathbf{s} \in \mathbb{R}^3$ and axis-angle $\mathbf{r} \in \mathbb{R}^3$, to represent the scaling and rotation matrix, respectively.

The 3D Gaussians are projected onto the 2D image plane during the rendering process. The resulting pixel color $C$ is computed by blending the $N$ projected 3D Gaussians primitives within that pixel. This process can be formulated as:

$$\mathbf{C} = \sum_{i=1}^{N} \alpha_i \prod_{j=1}^{i-1} \left(1 - \alpha_j\right) \mathbf{c}_i, \qquad (3)$$

where $c_i$ denotes the color of the $i$-th projected 3D Gaussians primitive, and $\alpha_i$ represents the blending weight calculated from the learned opacity and probability density.

## 4 METHOD

In this work, we propose $E^3$Gen, a generative model designed for efficient, expressive, editable digital avatar generation. An overview of $E^3$Gen is illustrated in Fig 2.

To achieve efficient, expressive and editable avatar generation, we propose a novel generative UV features plane representation (Sec 4.1). This representation ensures compatibility between the 3D Gaussian and the generative diffusion model while preserving efficiency. In Sec 4.2, we present the part-aware deformation module. This module offers full-body pose control, including facial expressions and gestures, allowing for expressive avatar animations. The training scheme of our method is detailed in Sec 4.3. Furthermore, in Sect 4.4, we discuss the editing capability of our method.

## 4.1 Generative UV Features Representation

Get inspiration from previous work for adopting 3D Gaussians in animatable avatar reconstruction tasks, we aim to introduce 3D Gaussians as a target space for diffusion model. Different from reconstruction tasks focusing on per-subject optimization, we have to enable 3D Gaussian training among multiple subjects, thus a shared structure among subjects is needed. To be compatible with the 2D-CNN-based denoising network in diffusion model, the shared structure is expected to be a 2D representation. Therefore, we propose to represent 3D Gaussian based digital avatars in the 2D UV space defined by the SMPL-X parametric model.

Given a generated UV features plane $\{x_i\}$, we extract a set of $K$ 3D Gaussian primitives from it to obtain a 3D Gaussian-based generated avatar $\mathcal{G}$:

$$\mathcal{G} = \{\mathcal{G}_k\}_{k=1}^{K}, \text{ where } \mathcal{G}_k = \{\mu_k, \alpha_k, \mathbf{r}_k, \mathbf{s}_k, \mathbf{c}_k\}. \qquad (4)$$

Each Gaussian primitive $\mathcal{G}_k$ is parameterized by a 3D position $\mu_k \in \mathbb{R}^3$, an opacity $\alpha_k \in \mathbb{R}$, a rotation matrix $R$ represented by the axis angle representation $r_k \in \mathbb{R}^3$, a scale matrix $S$ represented by a diagonal vector $s_k \in \mathbb{R}^3$, and a rgb color $c_k \in \mathbb{R}^3$.

The input generated UV features plane is first separated evenly into two parts and then decoded with two light-weight shared decoders $D_a$ and $D_g$, respectively. The separation of the UV features plane enables the disentanglement of geometry and texture, facilitating capability in editing. $D_g$ predicts geometry related attributes of 3D Gaussians: position $\mu$ and opacity $\alpha$, while $D_a$ predict appearance related attributes of 3D Gaussians: scale $\mathbf{s}$, rotation $\mathbf{r}$ and color $c$. As demonstrated by Li $et.al$ [33], convolutional-based decoder provide realistic results with more details than MLP-based decoder. Thus, we construct $D_g$ and $D_a$ as shallow convolutional-based networks for fast inference and high generation quality.

With these decoded attribute maps, we can extract a 3D Gaussians-based avatar in canonical pose space. Specifically, we query the attributes of those Gaussian primitives $\mathcal{G}_k$ by projecting them according to their initial position $\hat{\mu}_k$ onto each of the five attribute planes, retrieving the corresponding attributes via bilinear interpolation, thus obtaining a set of Gaussian primitives $\mathcal{G}$ which represent a digital avatar. In practice, the position $\mu_k$, scale $\mathbf{s}_k$ and rotation $\mathbf{r}_k$ of one Gaussian primitive $\mathcal{G}_k$ are modeled relative to the SMPL-X template as follows:

$$\begin{aligned} \mu_k &= \hat{\mu}_k + \delta_{\mu_k} \\ \mathbf{s}_k &= \hat{\mathbf{s}}_k \cdot \delta_{\mathbf{s}_k} \\ \mathbf{r}_k &= \hat{\mathbf{r}}_k \cdot \delta_{\mathbf{r}_k}, \end{aligned} \qquad (5)$$

where $\hat{\mu}_k$, $\hat{\mathbf{s}}_k$ and $\hat{\mathbf{r}}_k$ are initial values based on SMPL-X parametric model. $\delta_{\mu_k}$, $\delta_{\mathbf{s}_k}$, $\delta_{\mathbf{r}_k}$ represent the predicted values queried from attribute maps. The weak constraints to the surface of SMPL-X model ensure reasonable generation results without undermining the representation ability of the original 3D Gaussians.

**Gaussian Primitives Initialization.** The positions $\hat{\mu}$ of Gaussian primitives are initialized by sampling the center points of faces on a densified SMPL-X model. Similar to TADA [34], we subdivide the SMPL-X model to enhance the generation quality. We do not sample points uniformly on the UV space as the UV space shows a sparsity of points on the body surface which might undermine the representation ability. All Gaussian primitives are assigned an

initial scale $\hat{s}$ according to the distances between them and their neighbors. The initial scale is calculated in the targeted pose space to enable stretching ability for reasonable animation results. Different from origin 3D Gaussians, we set the orientation of each Gaussian primitive as the local tangent frame of the 3D surface point, similar to Lombardi *et.al* [36]. This initialization introduces human prior which alleviates the spiking artifacts during the animation process. We also adopt a different rotation representation: axis-angles compared to the quaternion utilized in previous work. Because the elements in this representation have the same value range which stabilizes the optimization of the neural network.

## 4.2 Part-aware Deformation Module

To achieve expressive full-body pose control, we propose a part-aware deformation module to transform the extracted digital avatar $\mathcal{G}$ into targeted pose space with accurate control over hands and face. The deformation module follows a forward skinning scheme based on linear blend skinning technique. Specifically, we apply the following transformation to the position $\mu_k$ of each Gaussian primitive $\mathcal{G}k$:

$$\mu'_k = \sum_{i=1}^{n_b} w_i \mathbf{B}_i \mu_{\mathbf{k}}, \tag{6}$$

where $n_b$ is the number of joints, and $\mathbf{B}_i$ denotes the transformation matrix of the $i$th joint from the canonical pose space to the targeted pose space. $w_i$ represents skinning weights, which determine the influence of the motion of each joint on position $\mu_k$ quantitatively.

To compute the skinning weights field for our deformation module, we leverage the predefined skinning weights on the SMPL-X mesh due to the complexity of diffusion training scheme compared to reconstruction tasks. Additionally, since the hands and face regions are relatively small but exhibit intricate deformation, full-body pose control has long been challenging. With the alignment to SMPL-X and the observation that the face and hands regions undergo minimal topology changes, we can compute the skinning weights directly based on their barycentric coordinates. This approach ensures accurate and robust animation for these specific regions. For body parts where large topology changes may occur, applying the same technique directly would result in artifacts. Drawing inspiration from AG3D [16], we represent the skinning weights field for body parts using a low-resolution volume. Each voxel in the volume is assigned skinning weights by calculating the values through the accumulation of skinning weights from each of the K nearest vertices on the densified SMPL-X surface, weighted by inverse distance. Gaussian primitives associated with the body part can then retrieve their skinning weights through trilinear interpolation from the volume. This approach enables accurate and smooth deformation while handling large topology changes effectively. Different from [16], our deformation method enables full-body animation, whereas AG3D focuses on specific body parts. Additionally, we avoid the costly root-finding process associated with the implicit representation adopted by AG3D, achieve more efficient deformation.

**3D Gaussian Attribute Deformation.** During the deformation process, the opacity $\alpha$ and color $c$ remain unchanged, while the scale $s$, rotation $r$ and position $\mu$ change. We have already discussed the deformation of position $\mu$ in the previous paragraphs. In this part, we try to tackle the deformation in rotation $r$ and scale $s$. The rotation $r$ is updated using the following formula:

$$\mathbf{R}' = \mathbf{T}_{1:3,1:3}\mathbf{R}, \text{ where } \mathbf{T} = \sum_{i=1}^{n_b} w_i(\mu)\mathbf{B}_i, \tag{7}$$

where $\mathbf{R}$ is the rotation matrix derived from the axis angle representation $r$, and $\mathbf{T}$ is the transformation matrix computed as the weighted sum of the bone transformations $\mathbf{B}_i$. $w_i(\mu)$ corresponds to the skinning weights associated with the position $\mu$. To account for the change in scale $s$, we define the initial value of $s$ in the targeted pose space. Specifically, using the deformed Gaussian primitives, we determine the initial value of $s$ based on the distance to its neighboring primitives.

By updating the rotation $r$ and scale $s$, we ensure that the deformations applied to the Gaussian primitives encompass changes in rotation, scale, and position, , allowing for robust and accurate pose control.

**Adaptation to Multi-subjects.** To be compatible with the generation task, which are trained with multiple subjects in various body shapes, we disentangle the body shape factor from. We model the generated avatar in canonical space with a neutral body shape. And add a warping process to map the neutral body avatar into the targeted body shape space before transforming to the targeted pose space. The warping process can be formulated as follows:

$$\bar{\mu}(\beta) = \mu + B_S(\beta, \mathcal{S}, \mu), \tag{8}$$

where, $\mu$ is the positions of 3D Gaussian primitives in canonical space. $B_S(\beta; \mathcal{S}, \mu)$ is the offset derived from the displacements on each SMPL-X vertex calculated by shape parameters $\beta$ with corresponding bases $\mathcal{S}$. To enable accurate expressive control and alleviate artifacts in joints, we further add expression offsets $B_E(\psi; \epsilon)$ and pose correction term $B_P(\theta; \mathcal{P})$ to the warped 3D Gaussians, follow SMPL-X deformation process.

## 4.3 Training

We follow the single-stage diffusion training process where UV features plane fitting and denoising processes are conducted simultaneously. The total training objective can be formulated as:

$$\mathcal{L} = \lambda_{\text{fit}} \, \mathcal{L}_{\text{fit}} \, (x_i, \psi) + \lambda_{\text{denois}} \, \mathcal{L}_{\text{denois}} \, (x_i, \phi), \tag{9}$$

where $x_i$ denotes the UV feature plane, $\phi$ and $\psi$ are parameters of the denoising U-Net and shared decoder, respectively. $\mathcal{L}_{\text{fit}}$ and $\mathcal{L}_{\text{denois}}$ represent training objective for the fitting and denoising process. $\lambda_*$ are loss weights. Previous diffusion-based methods adopt a two-stage training scheme, where a fitting process is trained first to obtain per-subject latent feature planes. The obtained latent feature planes are utilized as ground truth for the second-step denoising process training. Different these methods, our UV feature planes $\{x_i\}$ are constrained by both terms in the loss function. The introducing of denoising process constrain for UV feature plane optimization is beneficial for learning unseen regions in the training data as demonstrated by previous work [6].

**Fitting Process.** During the fitting process, we optimize the UV features plane as well as the geometry and appearance decoder via the following loss function:

$$\mathcal{L}_{\text{fit}} \, (x_i, \psi) = \lambda_{\text{c}} \mathcal{L}_{\text{c}} + \lambda_{\text{vgg}} \mathcal{L}_{\text{vgg}} + \lambda_{reg} \mathcal{L}_{\text{reg}}. \tag{10}$$

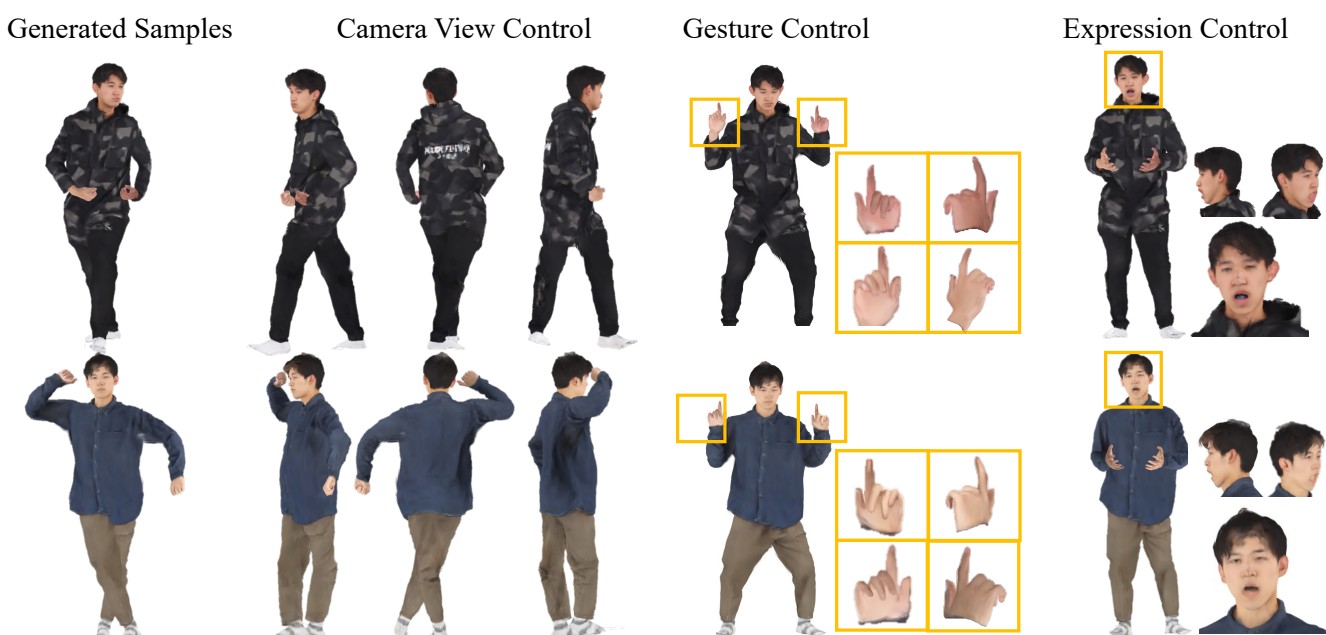

**Figure 3: We demonstrate the effectiveness of our method in achieving precise and robust control over facial expressions and gestures. Our approach enables clear and distinct control over each individual finger, ensuring their visibility and accurate positioning. Additionally, our method exhibits strong robustness when faced with novel poses, producing reasonable and plausible results for facial expressions.**

The color loss $\mathcal{L}_\text{c}$ computes the L2 distance between the ground truth images and the rendered results of our generated avatars. We randomly sample several images instead of one from all available observations for each scene in one training step to prevent the model from getting stuck in a local minimum. Different from SSD-NeRF [6] which is limited by implicit representation that can only be optimized via per-pixel objectives, the efficiency of generative UV features representation enables us to render the whole images and apply perceptual loss [25] on them. Specifically, the perceptual loss is calculated based on the features maps of ground truth images and our rendered outputs which are extracted from a pre-trained VGG [53] network. $\mathcal{L}_\text{reg} = \|\delta_{\mu_k}\|_2^2$ constrains the predicted offset values from being unreasonably large.

**Denoising Process.** For digital avatar generation, we utilize the diffusion model to learn a mapping from Gaussian noise to generative UV features plane. With Gaussian noise as input, the diffusion model can denoise it and output a reasonable UV features plane. During this process, we optimize both the generative UV features plane $x_i$ and the denoising UNet's parameters $\phi$. Specifically, we first add Gaussian noise $\epsilon \sim \mathcal{N}(0, I)$ into the given generative UV features plane $x_i$ via a noise schedule comprising differentiable functions $\alpha(t)$ and $\sigma(t)$, obtaining a noisy feature plane $x_i(t) := \alpha(t)x_i + \sigma(t)\epsilon$ at diffusion time step $t$. Then, we utilize the denoising UNet to obtain the denoised output $\hat{x}_i$ via:

$$\hat{x}_i = \alpha(t)x_i(t) - \sigma(t)\hat{v}, \tag{11}$$

where $\hat{v} \equiv \alpha(t)\epsilon - \sigma(t)x_i$ according to the $v-$parameterization method proposed in [48]. The denoising loss is formulated as following:

$$\mathcal{L}_\text{denois}(x_i, \phi) = \mathbb{E}_{i,t,\epsilon}\left[\frac{1}{2}w(t)\|\hat{x}_i - x_i\|^2\right]$$
$$w(t) = (\alpha(t)/\sigma(t))^{2\omega}, \tag{12}$$

where $t \sim \mathcal{U}(0, T)$, $\omega$ is a hyperparameter which we empirically set to 0.5.

## 4.4 Editing

Our novel representation, generative UV features plane, facilitates various customization applications, including local region editing and attribute transfer between subjects. By disentangling geometry and appearance, we expand the capabilities for editing, allowing for editing on either geometry or appearance individually. We provide visual examples in our experiment results(Fig 6).

**Local Region Editing.** In contrast to previous methods that employ monolithic representations, where generated avatars are treated as a unified entity with entangled attributes, the generative UV features plane represents 3D digital avatars as a collection of 3D Gaussian primitives that are loosely connected to the SMPL-X parametric model. This representation allows for enhanced flexibility and freedom in editing the avatars. Each Gaussian primitive can be independently modified by optimizing its associated geometry or appearance UV features, or by directly manipulating its attribute values.

**Attribute Transfer.** Benefiting from the shared structure among subjects, we can easily transfer geometry and appearance attributes

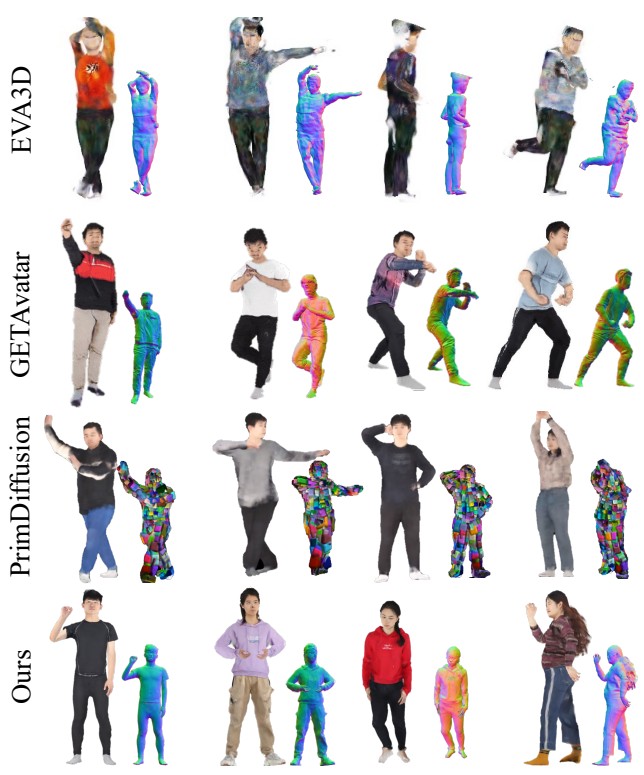

**Figure 4: Qualitative Comparison. Our method demonstrates superior performance in rendering quality and geometry quality compared to other methods. Due to the challenge of obtaining normals directly from PrimDiffusion, we visualize its mixture primitives as a rough representation of the geometric structure.**

from one generated subject to another by swapping the corresponding features. The utilization of UV space, which ensures semantic consistency, further facilitates the transfer of specific regions such as the nose, clothing, and face.

# 5 EXPERIMENTS

**Dataset.** In our experiments, we utilized the THUman2.0 Dataset [60] as our training data. This dataset comprises 526 textured 3D scans captured using a dense DSLR rig, covering a wide range of poses. Each scan is accompanied by its corresponding SMPLX parameters. Our data pre-processing involves rendering 500 identities from the THUman2.0 dataset using 54 camera views for each identity. Notably, our training did not rely on explicit 3D supervision, such as normals or 3D meshes, ensuring future extension to multi-view video datasets.

**Metrics.** Consistent with previous works, we evaluate the quality and diversity of the generated avatars using the Fre'chet Inception Distance (FID) metrics. To compute these metrics, we employ a set of 50,000 rendered multi-view images. Additionally, we extend our analysis by calculating the $FID_{norm}$ metrics between the generated and ground truth normal maps, providing insights into the quality

**Table 1: Quantitative comparison on the THUman2.0 [60] dataset, and the ∗ results are adopted from GETAvatar [63].**

| Methods | Res | FID↓ | $FID_{norm}$ ↓ | FPS↑ |
|---|---|---|---|---|
| ENARF* [40] | 128 | 124.61 | 223.72 | 8 |
| GNARF* [3] | 256 | 68.31 | 166.62 | 8 |
| EVA3D [23] | 512 | 60.82 | 60.67 | 6 |
| GETAvatar* [63] | 1024 | 17.91 | 55.02 | 17 |
| PrimDiffusion [11] | 512 | 62.43 | NA | 88 |
| Ours | 1024 | **15.78** | **25.63** | **110** |

of generated geometry. We estimate the normal values based on the axis directions of each 3D Gaussian Primitives.

## 5.1 Evaluation of Generated Avatars

*5.1.1 Generation and Animation Capacities.* **Random Generation.** We demonstrate the capability of our method to generate diverse and detailed avatars with controlled actions and camera angles in Fig 1 and Fig 3. Our method ensures view consistency and produces high-resolution rendering results for the generated avatars, even with a given camera pose. Additionally, we achieve accurate and robust control over hand and facial expressions. Notably, despite being trained on the THUman2.0 dataset, which has limited facial expressions, our method is able to generate reasonable open mouth expressions that are not explicitly present in the dataset. These results highlight the effectiveness and versatility of our approach.

*5.1.2 Comparisons.* We compare our method with representative approaches in both implicit representation and explicit representation, including those utilizing 3D Aware GANs or diffusion models. The comparison results are presented in Table 1. The visual results in shown in Fig 4. Our method demonstrates superior visual quality and diversity and achieves 100FPS rendering speed for high-resolution rendering, as measured by FID and FPS. This demonstrates the power of our proposed generative UV features plane. Furthermore, even without the supervision of normal maps and 3D models, our method achieves high performance in capturing detailed geometry surfaces, as indicated by the $FID_{norm}$ metrics. Constrained by implicit representations, GNARF [3], ENARF-GAN [40], and EVA3D [23] exhibit limited frames per second. GETAvatar [63] achieves faster rendering benefiting from its explicit Mesh representation. PrimDiffusion, which adopts a mixture of primitive representation, achieves an extraordinary FPS of 88. However, obtaining geometry surfaces from this representation is challenging. The results tend to be blurry, we guess that might result from artifacts caused by the overlapping patches.

## 5.2 Ablation Study

**Generative UV features VS Generative UV attributes.** Instead of intuitively producing the Gaussian Attribute maps, we encode them as latent feature code planes, which enhance the training stability and lead to significant generation quality improvements according to Tab 2. This is due to the complexity of the attributes of 3D Gaussians, which makes them challenging to optimize directly. To address this issue, we employ a lightweight decoder to preserve

**Table 2: Ablation study on the initialization of Gaussian primitives and the utilization of UV features.**

| Methods | FID↓ | FID$_{norm}$ ↓ | KID ↓ |
|---|---|---|---|
| w.o init | 16.74 | 51.42 | 13.75 |
| UV Attributes | 17.92 | 43.23 | 15.41 |
| Full Pipeline | **15.78** | **25.63** | **13.30** |

the efficiency of the original 3D Gaussians. As shown in the Tab 1, our model's speed is not significantly affected by the additional latent decoding process.

**The initialization of 3D Gaussian Primitives.** We initialize our generative UV features plane based on SMPL-X prior. Experimental results presented in Tab 2 demonstrate that this initialization approach yields improved generation quality and better surface geometry. The weak constraint to SMPL-X does not compromise the expressive power of our representation. As evident from the Fig 1, our method is capable of generating loose-fitting clothing that is not constrained by the SMPL-X topology.

**The part-aware deformation methods VS KNN-based forward skinning.** Due to the small facial area and its rich range of motion, using a K-nearest neighbors (KNN) based forward skinning method leads to errors, as depicted in Fig 5. The KNN-based forward skinning method fails to accurately open the mouth of the avatar according to the driving pose. In contrast, our part-aware deformation module takes advantage of the minimal topology changes in the face and hands, employing different approaches to obtain skinning weights for the body, face, and hands, resulting in robust and accurate full-body expressive pose control. Fig 3 demonstrates the precise control achieved over the fingers and facial expressions using our method. It is important to note that since our method does not model the interior of the mouth, the color display after opening the mouth represents the colors from the hair and neck region, and not artifacts caused by the driving method.

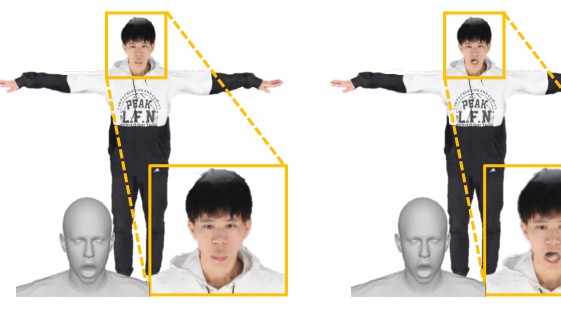

Forward Skinning KNN    Ours

**Figure 5: Ablation on deformation method. Our method achieves more accurate results for a given facial expression compared to the K-nearest neighbors (KNN) based forward skinning method.**

## 5.3 Applications

The Generative UV features plane supports editing of generated avatars, including local editing and attribute transfer between subjects. Fig 6 provides examples of these two editing methods. For local editing, the UV features plane represents a person as a collection of 3D Gaussian primitives. By modifying the attributes of local regions' 3D Gaussian primitives, local editing can be achieved. Fig 6 shows an example of editing the nose of the person is shown, resulting in a change in nose length. The edited avatar can still be controlled in a similar manner.

Due to the shared UV structure provided by the UV features plane among subjects, attribute transfer between subjects can be easily performed. The second row of Fig 6 demonstrates the effect of exchanging facial attributes. The first two columns show the original generated avatars, while the last two columns show the results after attribute exchange. This support for editing enhances the practical applicability of our method in industrial settings. For more editing results, please refer to the Sup. Mat.

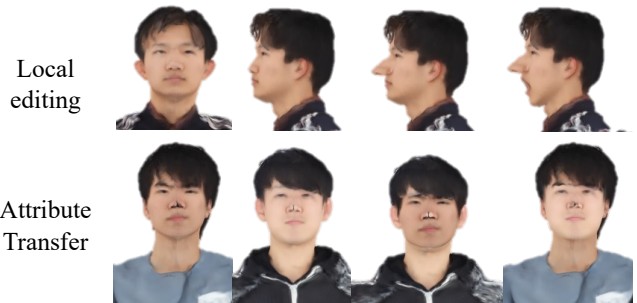

Local editing

Attribute Transfer

**Figure 6: Our method enables local editing and attribute transfer. In row one, we demonstrate the capability to modify only the nose of the avatar. The shared structure of UV featuers plane allows us to transfer attributes between different subjects, as showcased in row two.**

## 6 CONCLUSION

In conclusion, this paper introduces a novel method, called $E^3$Gen, for efficient, expressive, and editable digital avatar generation using 3D Gaussians. The paper addresses two major challenges in this task: the unstructured nature of 3D Gaussians and the animation of 3D Gaussians in a generative setting involving multiple subjects. To overcome these challenges, the proposed method introduces a generative UV features representation that encodes unstructured 3D Gaussians onto a structured 2D UV space defined by the SMPL-X parametric model. This representation preserves the expressive power of 3D Gaussians while introducing a shared structure among subjects, enabling generative learning of the diffusion model. To achieve robust and accurate full-body expressive pose control, a part-aware deformation module is proposed. This module enables precise control and editing of avatar poses. Extensive experiments demonstrate the superior performance of the proposed method in avatar generation, as well as its ability to enable expressive full-body pose control and editing.

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
