# OpenReview forum: "$E^{3}$Gen: Efficient, Expressive and Editable Avatars Generation"
_acmmm.org/ACMMM/2024/Conference — MM2024 Poster_

### Official Review · Reviewer_U1MH · 2024-05-07

**Rating:** 4
**Confidence:** 2

**Summary:**

This paper focues on generative 3D human avatars. It uses the 3D Gaussian Splatting (3DGS) as its representation. But instead of generating in 3D space, this paper propose to use 2D diffusion model to generate a UV feature plane, and employ two decoders to decode the feature plane as the properties of 3D Gaussian primatives. After generating the avatar, it could be animated via a part-aware deformation method which is based on SMPL-X.

**Strengths:**

I am not very familiar with 3D generative human avatars, so I may not be able to properly evaluate the novelty of this method. The strength of this paper is that it appears to be a relatively complete work. The generation and animation results in paper and supplementary video looks good. Compared with previous method, the avatar generated by proposed method are more realistic, and it enables animation of intricate details such as fingers.

**Limitations:**

1. I am not quite clear how the mapping between 2D UV and 3D avatar is determined. In particular, I would like to know how to determine the 3D position $\mu_k$ at a specific UV coordinate on the feature plane. Is it follow the uv of SMPL-X model? Additionally, in L418-L422, I wonder what is the shape of $x_i$, and how $K$ primitives could be extracted from $x_i$.

2. I also not understand why $x_i$ also need to be optimized in Sec 4.3. It is clear that the parameter of diffusion model $\phi$ should be optimized for generative modeling. However, it seems that $x_i$ is the ground truth UV feature map according to Eqn. 12, which appears strange if it is also optimized. Again, I believe the author should provide a more concrete explanation on what is $x_i$ and how it is constructed.

3. Another concern is that I wonder why the qualitative result of previous methods, in particular EVA3D [23] appears worse than the result presented in original paper.

Overall, the presentation current version is relatively unclear. I expect author give more explaination, and I will consider adjusting my score according to author's response other reviewer's opinion.

**Suitability:**

2

---

### Official Review · Reviewer_9cXV · 2024-05-24

**Rating:** 3
**Confidence:** 3

**Summary:**

This paper aims to introduce 3D Gaussians for efficient, expressive, and editable digital avatar generation. They propose an avatar generation method named E3Gen which use generative UV features representation to decode GS attributes and a part-aware deformation module to achieve  accurate full-body expressive pose control. Extensive experiments demonstrate the effectiveness.

**Strengths:**

1. Good writing and illustration.
2. A bunch of experiments are provided.

**Limitations:**

1. In the caption of Fig. 2, 'xi' should be latex equation $x_i$.
2. The reported results of baseline methods seem to be much worse than the original paper. Did you reproduce it correctly?
3. The novelty is limited. Both UV-Map and deformation are not new things.
4. From the visual cases, the nose part showcasing significant artifacts. Please explain this.
5. Why only 1024x results are reported in Tab. 1? What about the results of 512x?

**Suitability:**

2

---

### Official Review · Reviewer_Gajy · 2024-05-27

**Rating:** 4
**Confidence:** 3

**Summary:**

This paper proposes an avatar generation framework, named E3Gen. It consists of a generative UV features representation and a part-aware deformation module. The generative UV features marry the unstructured 3D Gaussians with the current diffusion generation pipeline. The part-aware deformation is designed correspondingly to hand, face region, and body parts for full body animation.

**Strengths:**

- This paper is well-written.
- The proposed generative UV features seem effective and may inspire further research.
- The part-aware deformation module allows for full-body animation, particularly enabling control of facial expressions and hand gestures, a capability not commonly present in previous works.
- The experiment shows superior generation quality.

**Limitations:**

- Densification and pruning in 3DGS are essential for high-fidelity reconstruction and generation. However, the number of Gaussians in E3Gen remains unchanged, potentially limiting its representational capacity. Exploring a Gaussian growth strategy for E2Gen would be interesting.

- The authors claim that compute the skinning weights based on their barycentric coordinates (a barycentric interpolation?) in hand and face reigns, which is not sufficient for reproduction. More details of the skinning weights computing are recommended to add.

- Some notable artifacts are noticed on the face region (e.g., a dark line/hole near the nose). Could the authors explain why this is the case and how it could be improved?

Typos: \
Line 251: xi -> x_i; \
Line 542: we disentangle the body shape factor from …\
Line 570: Different these methods -> Different with these methods

**Suitability:**

3

---

### Official Review · Reviewer_t2aX · 2024-06-04

**Rating:** 4
**Confidence:** 2

**Summary:**

This paper proposes E3Gen, a method based on gaussian splatting and diffusion model for generating animatable avatars. E3Gen trains a diffusion model to map noise to UV features, which are then decoded into Gaussian attributes in canonical space and subsequently warped into the target pose space. The entire training process is conducted end-to-end.

**Strengths:**

- Instead of performing denoising diffusion on the image feature space, E3Gen introduces a novel approach by learning the denoising process in the UV feature space.
- EG3D enables expressive animation, and the provided examples for hand and mouse animations look good.
- The writing is clear and easy to read.

**Limitations:**

- The animation might not be accurate, especially for bodies with large deformation. This is because E3Gen heavily depends on SMPL-X skinning weights. TADA works well for animation because it models large deformations using shape parameters, while small deformations are handled by vertex offsets. However, in this paper, the shape parameters are fixed, which means the difference between the SMPL-X body and clothed body are entirely managed by the offsets. If the offsets are large, the deformation would be unreliable.
- There is no evaluation on real image examples. It would be beneficial if the authors could provide reconstruction and animation visualization results on real images.
- There are limite results for facial animation. I am curisous about the perfomance, particularly how consistent geometry and texture are during facial animation,  especially for the mouse?

**Suitability:**

2

---

### Meta-Review · Area_Chair_S7bY · 2024-06-30

**Recommendation:** Accept (Poster)
**Confidence:** 4

**Metareview:**

The paper received mixed scores, with a positive tendency (BA,BR,BA,BA). The reviewer suggesting BR provided a superficial and unclear review, with a very few details and did not provide a final rating after rebuttal. Other reviewers provided more detailed comments.

The positive aspects are:
- E3Gen introduces a novel approach by learning the denoising process in the UV feature space which seems effective and may inspire further research.
- Results show improved performance, and some specific new abilities such as modeling fine details such as fingers motions.

The rebuttal clarified most of the issues raised by reviewers. They were mostly related to unclear technical details, defects in the generated samples or minor tests missing.

Overall, both the reviewers comments, the results from the paper and the interesting technical design point in favor of the paper acceptance.